# Moral panic about "covidiots" in Canadian newspaper coverage of COVID-19

Gabriela Capurro[1], Cynthia G. Jardine[2], Jordan Tustin[3], Michelle Driedger[1]*

**1** Department of Community Health Sciences, University of Manitoba, Winnipeg, Manitoba, Canada,
**2** Faculty of Health Sciences, University of the Fraser Valley, Chilliwack, British Columbia, Canada, **3** School of Occupational and Public Health, Ryerson University, Toronto, Ontario, Canada

* michelle.driedger@umanitoba.ca

## Abstract

Moral panics are moments of intense and widespread public concern about a specific group, whose behaviour is deemed a moral threat to the collective. We examined public health guidelines in the first months of the COVID-19 pandemic in Canadian newspaper editorials, columns and letters to the editor, to evaluate how perceived threats to public interests were expressed and amplified through claims-making processes. Normalization of infection control behaviours has led to a moral panic about lack of compliance with preventive measures, which is expressed in opinion discourse. Following public health guidelines was construed as a moral imperative and a civic duty, while those who failed to comply with these guidelines were stigmatized, shamed as "covidiots," and discursively constructed as a threat to public health and moral order. Unlike other moral panics in which there is social consensus about what needs to be done, Canadian commentators presented a variety of possible solutions, opening a debate around infection surveillance, privacy, trust, and punishment. Public health communication messaging needs to be clear, to both facilitate compliance and provide the material conditions necessary to promote infection prevention behaviour, and reduce the stigmatization of certain groups and hostile reactions towards them.

## Introduction

In December 2019, the world learned about a novel coronavirus that was spreading throughout the Chinese province of Wuhan. In a matter of weeks, the SARS-CoV-2 virus spread across the globe, causing the worst pandemic in over a century and leading countries to enact mandatory quarantines and infection prevention regulations. At the time of writing, COVID-19, the infection caused by the novel coronavirus, has infected over 99 million people worldwide and caused more than 2 million deaths [1]. Several infection prevention and control measures have been adopted in different jurisdictions across Canada, from mandatory use of facemasks to the banning of indoor gatherings and keeping two meters of physical distance from others at all times. These public health guidelines have reshaped our social mores, normalizing behaviours previously considered antisocial, and causing anxiety or concern over those who refuse to comply with them.

**Data Availability Statement:** Details regarding the minimal data set are included in the Methods of the paper.

**Funding:** M.D., C.J., and J.T. are recipients of a CIHR grant (OV6 – 170370) Canadian Institutes of

Health Research (https://cihr-irsc.gc.ca/e/193.
html) The funders had no role in study design, data
collection and analysis, decision to publish, or
preparation of the manuscript.

**Competing interests:** The authors have declared
that no competing interests exist.

The imposition of various provincial and federal guidelines to control the spread of
COVID-19 in Canada were met with criticism, with some arguing these measures should have
been adopted earlier or should be more stringent, while others rejected them as a violation of
civil liberties. Compounding these tensions is the foundation of Canada's confederation,
where the division of powers and responsibilities over public health and health service delivery
vary drastically. In addition, the primary spokesperson for communicating the status of the
disease and control measures differed between jurisdictions. The variability of COVID-19
prevalence, and concerted efforts to not impose a one-size-fits-all approach, resulted in differ-
ent public health control measures being invoked in different provinces and territories, and
indeed within different regions within a given jurisdiction, over time. We examined newspa-
per coverage of COVID-19 to assess how these public health guidelines have been discussed in
editorials, columns, and letters to the editor. We argue that the normalization of infection pre-
vention behaviours led to a moral panic which is expressed in opinion pieces through a
claims-making process in which compliance with public health guidelines is construed as a
moral imperative and a civic duty. Those who fail to comply with these guidelines were
shamed and discursively censured as a threat to public health and moral order.

## Moral panics

Moral panics are episodes of intense concern about the behaviour of a group or a particular
event. These moments tend to be volatile, as they emerge suddenly and then quickly dissipate
[2].This concern usually involves some hostility against those considered at fault and a dispro-
portionate depiction of the threat, as well as a broad and consensual reaction [3]. During
moral panics, deviant groups are considered "a threat to societal values and interests" [3].
These groups are identified and perceived as 'folk devils' [2], marginalized groups that embody
the social anxieties of the dominant group [4]. Moral panics, however, are not random and
irrational episodes, but are rather moments of power struggle between various asymmetrical
interests [5].

News media play a central role in the construction of moral panics [6]. Social reaction to
the contentious issue can be more or less consensual, and news media reproduce both moral
discourses and counter-discourses. They inform what constitutes immoral and deviant behav-
iour, as various social actors engage in claims-making processes, including so-called 'folk dev-
ils' [4, 6]. Therefore, the feeling of threat is further magnified in media discourse through
stereotypical representations of deviants and the expression of moral outrage by righteous fig-
ures and experts [2].

Some moral panics dissipate without long lasting effects, while others translate into policy
changes that remain after the panic has subsided [cf. 7, 8]. Moral panics also vary in intensity,
duration, and social impact depending on the extent to which the discourse resonates with
wider sociocultural anxieties [3]. For example, news coverage of outbreaks of vaccine-prevent-
able diseases tend to focus on 'anti-vaxxers,' a derogatory term used to refer to those who
oppose vaccination, despite the many degrees and reasons for vaccine hesitancy [9]. Within
such news coverage, 'anti-vaxxers' are portrayed as folk devils that endanger the community,
exaggerating the actual proportion of unvaccinated people in a population, and demanding
the punishment of those who ignore their 'moral duty' of vaccinating themselves and their
children for the protection of others [10].

Ungar [11] refers to moments of moral panic during health crises as "viral moral panics,"
which are often localized phenomena that involve the use of morality to regulate public behav-
iours. Moral obligations can include self-isolating when symptomatic, conforming to quaran-
tine regulations, and getting vaccinated when recommended. Furthermore, these episodes of

widespread attention and fear become part of the collective memory and are evoked whenever a new outbreak occurs. For example, the 1918 flu pandemic is frequently evoked as a metaphor of the worst-case pandemic scenario, a trope that has been used in media coverage of recent outbreaks (such as pandemic H1N1 influenza and Middle East Respiratory Syndrome or MERS), and resulting in exaggerated claims that did not materialize [11].

## Media coverage of health crises

Moral panics generated by diseases are common, as some medical conditions are imbued with stigma or become racialized [12], for example syphilis in the 19th century, HIV/AIDS in the 1980s, and SARS in 2003. Health crises, such as pandemics, provoke anxiety and feelings of dread about particular groups of people and specific behaviours [13, 14], which are then reproduced and reinforced in media coverage [12, 15]. The general public usually learns about novel health risks, such as COVID-19, through news media [16], which also play a key role in the formation of moral panics by amplifying the feelings of threat and fear that some groups represent to others [6]. Media narratives stereotype and misidentify deviance, which, in the case of health panics in particular, can produce socially harmful representations that stigmatize certain groups [12]. Opinion discourse (as expressed in columns, op-eds, and editorials), in particular, offers readers a "distinctive and authoritative voice that will speak to them directly, in the face of troubling or problematic circumstances" [17].

Canadian news media has contributed to health panics by reinforcing the stigmatization and discrimination of specific groups. In Canada, news coverage of the 2003 SARS outbreak depicted Chinese Canadians as both a biomedical and moral risk by associating this group with the origin of the outbreak in Hong Kong and directing fears and anxieties against them [18, 19]. SARS quickly caused a moral panic and people who were infected were stigmatized and blamed for endangering others [20]. Moral panics about epidemics and pandemics can extend into social media platforms as news stories, magnifying the threat posed by certain groups are shared thousands of times [21].

The first case of COVID-19 in Canada was detected in a Toronto hospital on January 25th, 2020 [22], evoking Toronto's experience as the epicentre of the 2003 SARS epidemic in Canada, which caused 43 deaths and over 400 suspected cases [23]. Since then, COVID-19 has been the leading news story every day in Canada, including guidelines and restrictions to minimize the spread of the virus, and asking individuals and organizations to rapidly shift professional and social practices. In the process, personal protective measures have been formulated and new social norms have emerged regarding what is appropriate contact with others. We examined opinion pieces published in Canadian newspapers to assess how these new social norms have been discussed and interpreted, and whether they have elicited the expression of a moral panic.

## Materials and methods

We conducted a claims-making analysis [24] of newspaper coverage of COVID-19 to assess how public health guidelines and non-compliance have been discussed in editorials, columns, and letters to the editor. More specifically, we sought to determine whether three specific characteristics of moral panics were present in opinion discourse: (1) moral concern about a group's behaviour; (2) hostility against individuals in that group, who are portrayed as deviants; and (3) moral entrepreneurs demanding punitive action. The presence of these three expressions of moral reprobation in opinion discourse would be indicative of a moral panic.

Due to the high volume of news coverage of the pandemic, we focused our analysis on six topics featuring high levels of scientific uncertainty and/or debate, and how journalists made

**Table 1. Data collection.**

| Topic | Dates covered | Keywords | Number of articles |
|---|---|---|---|
| Travel Quarantine Isolation | March 5th- March 31st | ["Travel"] and ["quarantine" or "isolation"] and ["Covid-19" or "Covid19" or "coronavirus"] | 222 |
| Epidemiological models | April 1st- April 30th | ["Model*" or "modelling" or "projection*"] and ["Covid-19" or "Covid19" or "coronavirus"] | 148 |
| Testing | May 5th- May 25th | ["Test*"] and ["Covid-19" or "Covid19" or "coronavirus"] | 106 |
| Face masks | March 1—July 14th | ["Mask*" or "face mask"] and ["Covid-19" or "Covid19" or "coronavirus"] | 374 |
| Physical distancing | March 20th–April 4th | ["Distancing" or "social distance" or "physical distance"] and ["Covid-19" or "Covid19" or "coronavirus"] | 255 |
| Airborne transmission | May 1st–July 15th | ["indoor transmission" or "aerosol*" or "airborne" or "microdroplet*"] and ["Covid-19" or "Covid19" or "coronavirus"] | 38 |

sense of them. In this article, we focus on three of these topics that contributed to the construction of a moral panic: (1) travel, isolation and quarantine; (2) use of facemasks; (3) and physical distancing.

Our sample for analysis included print and online coverage in seven Canadian newspapers, including both hard news and opinion discourse (editorial, op-ed): two with national distribution (*The Globe and Mail* and the *National Post*) and five with a more regional focus (*Montreal Gazette*, *The Toronto Star*, *Ottawa Citizen*, *Winnipeg Free Press*, and *Vancouver Sun*). We searched the database Factiva using various combinations of the keywords "COVID-19" "Covid19" "coronavirus" "quarantine" "isolation" "travel" "model*" "guideline" "test*" "mask*" "aerosol*" "airborne" "distancing" occurring between March and mid-July, 2020. We ended data collection for a specific topic at logical moments when news coverage of the topics waned in favour of new issues gaining greater salience. The sample for each topic encompassed at least 2 weeks of coverage to a maximum of 10 weeks (see Table 1). News articles that mentioned these topics only tangentially were not considered for analysis. The sample was composed of 1143 articles. We comply with all Terms and Services required by Factiva and necessary permissions for use.

The relevant articles were uploaded to NVivo12 for open coding analysis. We developed an initial codebook with five main codes (uncertainty; evidence; shifting guidelines; disagreement among experts; and trust) while iteratively adding new code categories as these emerged during the analysis. The stories were also coded for type of story (hard news, opinion, letter to the editor, or other), geographical focus (local, regional, national, international), and whether or not the reporter was a specialist science/health journalist. A second level of analysis was then conducted on the coded data to identify whether the three elements of a moral panic (explained above) were present.

The sample for each topic was established according to its salience in news coverage over time, acknowledging both overlap in time between topics as well as differences in how much coverage specific topics received. We coded one topic at a time, to ensure the coding process was focused on how public health guidelines were discussed (or not) in each specific case. The lead author coded the entire sample and the senior author coded a subset of news articles to ensure coding agreement. Disagreements were resolved through discussion. For the sample subset where double coding was completed, the final overall Cohen's kappa coefficient for intercoder reliability was .97.

## Findings

We identified three moments in the news coverage of COVID-19 in which columnists and commentators expressed concerns that suggest a moral panic about infection prevention

behaviour. First, they identified the desirable behaviour as a moral duty; second, those who failed to comply with public health guidelines were shamed and represented as 'folk devils'; and third, punishment for deviants was demanded in opinion pieces. Within these stages, however, there were some alternative narratives focusing on political responsibility and leadership versus an overly simplified view, characteristic of a moral panic, that emphasizes the role of individual behaviour.

## Self-regulation as moral duty

Complying with physical distancing and staying at home except to get food and medicines were the first guidelines that Canadians received from public health authorities on March 15th, 2020. Columnists and commentators reflected on this and other public health directives as they were announced in the following weeks and months, often presenting these actions as necessary. In opinion pieces, stopping the spread of COVID-19 was discussed in terms of individual responsibility for the common good, with references to Prime Minister Trudeau's pleas for Canadians to stay home as "a call to war"' and "a national duty". For example, an editorial in the *Globe and Mail* on March 23rd referred to the public health response to COVID-19 in military terms, calling it "a generational call to national duty; a request to do our part in the war on the coronavirus" [25]. A similar opinion was expressed the following day by a columnist in the *Winnipeg Free Press* who referred to compliance with public health guidelines as a civic duty and spoke of the role citizens have in stopping the spread of COVID-19, highlighting that "individuals have to do their part by staying home, practicing social distancing, and not hanging around in groups" [26]. Another columnist in the *Vancouver Sun* claimed on March 20th that public health authorities were asking Canadians "to make a few sacrifices for the greater good" and to be "our best selves, good citizens and neighbours[. . .] even in long grocery lineups" [27].

In the opinion pieces, there was constant acknowledgement of the important cultural shift that public health measures were demanding. Therefore, adapting individual behaviour was considered not only good: it was portrayed as a selfless sacrifice for the protection of the wider community. Columnists claimed that staying away from friends and family, and even strangers, goes against our cultural norms and how we have been socialized, while acknowledging the pandemic has changed our cultural understandings of how friendliness and politeness should be expressed. A columnist in the *Toronto Star* reflected on the challenge this cultural change imposes in the context of people who have been denounced in news media and social media for violating public health restrictions:

> This follows the closing of national and provincial parks and various conservation areas across Ontario after photos and videos of irresponsible people clumping together last weekend were circulated. Some of those clumpers were probably willful idiots, some just ignorant, but I've being thinking about how difficult it is, conceptually, to practise physical distancing. Walking away from and far around people goes against everything we've been taught. It feels so rude, though it's now the ultimate show of social respect and trust [28].

In this opinion piece, the columnist points to an important cultural shift that occurred in a matter of weeks, claiming that keeping physical distance is no longer unfriendly but has become an expression of consideration and respect. Therefore, those who still attend crowded events are either "ignorant" or "idiots," but above all they are described as "irresponsible." Similarly, health columnist André Picard noted this in the *Globe and Mail* and highlighted the symbolic meaning that facemasks have acquired:

The evidence for mask-wearing as a means of infectious disease prevention remains mixed at best. Increasingly, however, making and wearing a mask in public is becoming a demonstration of civic-mindedness, a public acknowledgment of the risks coronavirus poses, and a fitting symbol for the new normal [29].

Preventing the spread of COVID-19 is therefore framed as an individual responsibility for the good of the collective, which writers claimed is a moral duty, an act of selflessness and a show of respect and kindness, "literally a matter of life and death" [30]. Those who do not respect the rules, either willfully or by ignorance, are considered irresponsible and immoral.

## Shaming transgressors: Covidiots, young adults and snowbirds

Frequently documented in columns, editorials and letters to the editor was the feeling that the burden of infection prevention and control measures is being carried unevenly in Canada. In these pieces there is a strong focus on transgressors, who are shamed as 'covidiots,' a label used to designate people who do not wear facemasks, do not keep physical distance, and do not self-isolate as instructed.

At the beginning of the pandemic, news coverage of self-isolation and physical distancing guidelines focused on the various challenges people faced in order to comply with guidance measures. For example, the difficult situation that self-isolation posed for people living alone, the challenges parents faced by keeping their children at home, as well as the risk that not having access to parks and open-air spaces all negatively contributed to everyone's mental health. But soon the tone of the coverage shifted from empathetic human-interest stories to news reports of people not complying with public health guidelines. This shift was also reflected in opinion pieces as writers expressed anger and frustration over people who did not respect restrictions.

For example, a columnist in the *Toronto Star* explained that "scenes of people converging on beaches and in parks raised the ire of many; giving rise to one more new COVID-19-related term–"covidiots" (it went viral on Twitter)" [31]. The columnist also noted that even Prime Minister Justin Trudeau's tone had changed from one of positive reinforcement to one of calling out and shaming those who do not comply with safe distancing guidelines: "'Enough is enough,' he said, finally addressing the covidiots amongst us. Those who fail to quarantine now risk being fined" (ibid). This was also noted in an editorial published in the *Winnipeg Free Press* on March 27th, which not only referred to travellers not complying with the quarantine but also claimed that violators do not care about the threat they pose to others:

Hello, snowbirds. Welcome back. Go directly home. Do not go shopping (. . .) The federal government, which had previously urged Canadians returning from abroad to self-isolate for two weeks, has now declared that quarantine mandatory (. . .) why the suddenly urgent tone? Because, apparently, too many Canadians don't get it (. . .) There are many who just don't seem to care, and are stopping in at stores when they get across the border. This must stop. Do not go shopping. Go directly home [32].

In this editorial, travellers crossing the border from the United States in cars and campers, also called "snowbirds" because they spend winter in warmer zones, are described as ignorant and selfish, as well as failing to respect their civil and moral duty. The editorial also refers to the federal government invoking the Quarantine Act, which mandates quarantines for travellers, and deems it "quite frankly, embarrassing" (ibid) that travellers would need to be coerced with legal action as opposed to self-isolating voluntarily.

A columnist in the *Toronto Star* sought to ridicule people who do not respect safe-distance guidelines by claiming that even a preschooler can understand the new pandemic rules: "my kindergarten-aged niece informed me via FaceTime that the news said there were to be no play dates so she was going to play with her siblings instead (. . .) It seems many adults are not being as compliant as my niece" [33]. Meanwhile another columnist in the *Montreal Gazette* reinforced the notion of personal responsibility by urging people to comply with public health guidelines for the greater good: "Don't be the idiot who ruins it for everyone" [34]. Columnists focused most of their criticism on young adults and snowbirds, who were qualified in a *Winnipeg Free Press* column as COVID-19 deniers:

> This angst generally produces two reactions: those who accept the reality of the pandemic and those who foolishly do not. Among a long list of deniers have been teenagers on spring break, frolicking on Florida beaches until local officials shut most of them down; the recent crowds of people in Vancouver parading up and down Stanley Park's popular seawall path with little regard for social distancing [35].

However, despite the paralyzing effect that an unprecedented pandemic can elicit, another columnist in the *Winnipeg Free Press* explained that most Canadians are complying with public health measures: "We track legitimate sources of information, listen to the experts and accept that the threat is real and not exaggerated" [36]. The columnist, however, also criticized those who keep on with their lives and ignore the severity of the risk:

> As of last week, Canadians were still flying out for winter vacations. College students from across the United States were flocking to spring-break resorts. And they are not alone (. . .) one in eight Canadians believes the pandemic is overblown and as a result, are less likely to wash their hands, maintain social distance and avoid large public gatherings (ibid).

Besides describing them as "idiots," "irresponsible," "reckless," "shameless," and "careless," the opinion piece writers also raised the issue of the potential threat that non-complying individuals can pose to others. Due to asymptomatic transmission, transgressors are construed in the opinion pieces as an invisible threat to public health: "one reckless individual can contaminate hundreds of doorknobs, handles, railings, buses, taxis and subway cars" [37]. In another column it was emphasized that public health officials "had been begging the public to practice social distancing" and that "the message had been crystal clear. . .[and] it is not a hard concept to grasp. Keep your distance" [30].

A similar view was expressed by a columnist in the *Winnipeg Free Press* who stated that "in the case of COVID-19 it only takes a few deniers to sustain the threat from this virus". The columnist then exhorted those deviant individuals to "take note: we need you to stop denying the obvious and get with the program. You are literally killing us" [36]. A similar point was made by a *Globe and Mail* editorialist on the same day, who stated that "it is unconscionable at this point to act as though the pandemic doesn't involve you, or to assume that your individual actions aren't risky (. . .) the virus could also continue to feast on the indifference of a small but significant number of people who consider themselves above the fray" [38].

Readers also expressed concern in letters to the editor regarding lack of compliance with public health guidelines, particularly physical distancing. In a letter to the editor in the *Globe and Mail* a reader wrote on March 23rd: "I believe the biggest challenge is the failure of a segment of society to follow the direction of our leaders, especially in regard to the clear and repeated guideline to adhere to social distancing" [39]. Letters to the editor expressed more frustration as the first weeks of the pandemic went by, for example, admitting to being "mad at

those who refuse to stay at home except to get necessities, or who are not social distancing" [40] or feeling anxious over people around them not complying with physical distancing in public spaces, such as walking "in the middle of the sidewalk" or not paying attention to their surroundings and "walking around aimlessly" [41]. In a letter to the editor in the *Vancouver Sun* a reader warned that "if we don't behave and follow the rules, this social isolation is going to go on for a long time" and asked if "maybe the police should be enforcing the rules all over the city. Do we want that?" [42].

The moral outrage and condemnation of people who do not comply with public health rules, from social distancing and wearing a mask, to the two-week isolation after travelling, was expressed in a column in the *National Post* on April 18[th], which expressed the sense of risk and panic the author feels whenever a stranger stands too close to him as everybody must be assumed to be risky, and refers to the 'new politeness' in which individuals are expected to keep their distance:

> I treat pedestrians as though they are radioactive, veering out of their way as if to avoid oncoming shrouds of toxins. And I expect other people to act the same way—because social distancing only works if everybody is on board. So, it seemed to me a flagrant transgression, hovering in my immediate periphery as we waited for our companions to do their shopping. And yet my reaction—a curt "excuse me" combined with a reproving glare—was itself received as transgression, indeed causing the culprit to roll his eyes and scoff [43].

Marsh also expressed surprise that "this stubbornness remains inexplicably common," for example people who walk "hand-in-hand on narrow sidewalks" or "mild acquaintances stop to chat while walking their dogs, clogging up park paths" or people who look with reprobation "when you move out of their way or pause to give them space." The columnist refers to these people as "anti-distancers" who, despite their flagrant violations, look at those who comply with public health guidelines as if "you're the one irrationally panicking".

The opinion pieces expressed outrage and moral condemnation of people who do not follow public health guidelines, and whose behaviour is deemed a moral and physical threat. Transgressors are stigmatized, whereas those who comply with prevention guidelines are considered "selfless," "civic-minded," and "good citizens."

## Demanding punishment for 'deviants'

After defining what is moral and immoral behaviour, and identifying a group of transgressors, many readers and commentators reflected on the urgency to act fast to contain the pandemic. Writers claimed that action needed to be taken as punishment for not following public health guidelines. Notably suggestions included arresting and fining violators. A column in the front page of the *Vancouver Sun* on March 23[rd] noted that:

> The opportunity to supress this pandemic is rapidly closing and we've been put on notice that we can do this the easier way. We can wash our hands, keep our distance and stay home. Or we can do it the harder way, with more sickness, more deaths, and police officers pulled away from more pressing duties to enforce those orders [44].

Columnists in the *Globe and Mail* and *Montreal Gazette* echoed this idea, arguing that "masks should be mandatory on trains and buses. Many of the world's major cities, including Singapore, Berlin, Rome and Bangkok, have already taken this step" [45], and that "Montreal police should ticket anyone not respecting social-distancing norms." [34].

A *National Post* columnist went even further, arguing that Canadians should forgo some of their privacy rights to do surveillance of returning travellers. The columnist expressed disappointment that when the Canadian government announced a mandatory 14-day self-isolation period for travellers returning from abroad, who would be punished with up to 6 months in prison and/or a substantive fine if they did not self-isolate, no surveillance measures were implemented to ensure travellers complied. The columnist argued for enhanced surveillance of travellers and the use of technology to ensure compliance with self-isolation, stating that the federal government "must unleash the Big Brother bazooka by deploying technology [. . .] laws, fines and punishments, already exist but aren't working, nor is public shaming. Tracking Canadians using technology will keep everyone honest" [37]. Eventually public health officers followed up with travellers in isolation through phone calls and physical checks.

A similar argument was made by a *Winnipeg Free Press* columnist who criticized authorities in Manitoba for not disclosing more location-based information about community transmission of the virus: "In normal times, their refusal is understandable for legal and ethical reasons. But everyone will agree these are not normal times" [46]. At the same time he rejected notions of making public the personal information of those infected, because the latter could create a counter effect by dissuading people from getting tested and might generally lead to public shaming and stigmatization.

Another columnist in the *Vancouver Sun* argued that "the burden of social distancing is not being shouldered equally by all Canadians" and reflected on the possibility and legality of sanctions: "Individuals who are in breach of isolation orders can be criminally penalized. Given the nature of this unprecedented national crisis, such measures are unlikely to violate the Charter of Rights and Freedoms" [47]. Another columnist speculated about the possibility of "hotlines or volunteer watchdogs to become generally accepted" by people "frustrated by self-isolation, indefinite school closures and juggling work-family at home" [48]. The columnist then concluded that citizens reporting their neighbours is not "too harsh a call to make. People should face consequences for breaking distancing directives". Similarly, a columnist in *The Globe and Mail*, referred to the idea of asking individuals to act as vigilantes and report their neighbours, but concluded that such a measure would erode trust in government and public health:

> It's hard to imagine a better way to poison the we're-all-in-this-together sense that governments are trying to cultivate than a 1–800 line where people snitch on their neighbour hosting a poker game (. . .) The task of getting large numbers to suddenly change their behaviour relies on social conscience and peer pressure [49].

The *Globe and Mail* also criticized in an editorial on May 14[th] the lack of rigorous tracking of travellers, stating that trusting that people will actually self-isolate for two weeks once in Canada "makes no sense." The editorial claimed that "there is no testing, no tracking and no obligation for travellers to monitor their health and provide updates to public health authorities. Overseas travelers landing in a Canadian airport can even get on a connecting domestic flight before self-isolating" [50].

The issue of individual responsibility, however, was reframed by some columnists who argued that lack of compliance with public health guidelines is the result of city and provincial governments not facilitating physical distancing and safe access to outdoor spaces. For example, a columnist complained on April 3[rd] that the city of Toronto was not considering the idea of closing traffic lanes to make more space for pedestrians to follow the 2-meter distance rule: "city staff demonstrated their trademark total lack of imagination in their rejection of an idea to close off two lanes of Toronto's Yonge street" [51]. Selley also referred to the crowded condition in which some people live in Toronto and the risk this poses to mental health during the pandemic:

Imagine being cooped up 23 hours a day with your kids in a stifling shoebox[. . .] To keep people in such circumstances indoors by force, to denounce them even for taking a walk, to shut down schoolyards where kids could ride their bikes and scooters in at least relative safety, is to risk mental and physical health outcomes that should certainly be weighed against the risks of COVID-19 itself.

In a different column, on March 26[th], Selley shifted the blame from travellers returning home to the government, claiming that self-isolation guidelines and regulations have not been clearly communicated to Canadians returning from abroad:

One sheet of paper tells people to "self-isolate," but defines it as "not having visitors, especially older adults or those with medical conditions." That misbegotten adverb, "especially," does nothing except weaken the advice. Nothing on the handout suggests anything as alarming as "don't even stock up on groceries or refill your prescriptions before you lock yourself inside [52].

In the *Winnipeg Free Press* a columnist also referred to unclear public health and official communication stating that it "has been a source of confusion and stress about what exactly 'social distancing' means- and, more importantly, how to do it right" [53]. Furthermore, while most writers expressed the need to act faster to stop the spread of COVID-19 there was no agreement regarding what should be done. This leads to the questioning of core Canadian values, such as privacy versus surveillance, law enforcement versus peer pressure, and 'snitching' versus trust.

## Discussion and conclusion

The COVID-19 pandemic is an unprecedented health crisis that prompted a rapid public health response in Canada, premised on a series of changes in individual behaviour and cultural practices. In this paper, we argue that the normalization of infection prevention behaviours that the COVID-19 pandemic triggered in Canada led to a moral panic over the threat that people who do not comply with public health guidelines pose to the rest of society. This cultural shift led to the framing of compliance with public health guidelines as a desirable, moral behaviour positioning the role of individual responsibility at the centre of the pandemic response: if everyone makes sacrifices then the collective can get through this crisis; but if only some fulfil their duty, then the collective is at physical and moral risk.

Transgressors were stigmatized through the use of various labels, such as "covidiots," "careless," "irresponsible," and "embarrassing"; conversely, those who comply with prevention guidelines are considered "selfless," "civic-minded," and "good citizens." The identification of 'folk devils' and their stigmatization is characteristic of moral panics [2, 4, 5]. The initial focus in the newspaper coverage was on 'snowbirds' and young people on spring break, but as the virus spread in the community, 'transgressors' became anyone not wearing face masks, washing their hands, covering their cough, and keeping two meters of physical distance. This heightened concern over groups of people not keeping physical distance, isolating, and wearing a mask led to the expression of anger, frustration, and fear in the opinion pieces. Some writers confessed feeling apprehensive when people who do not keep distance around them at the grocery store or the park, while others expressed anger over the risk that these transgressors pose to others. These representations can lead to increased fear and feeling of risk as well as intensified hostility against those who fail to comply with COVID-19-prevention guidelines.

This inability to identify one well-defined group of 'deviants' increases the feeling of risk around others, an effect also found in the stigmatization of vaccine-hesitant parents as "anti-

vaxxers" or "science deniers" [cf. 10]. Additionally, shaming and ridiculing those who do not follow preventive measures for COVID-19 could lead–as in the debate around vaccination–to oversimplifying the reasons for these transgressions. But, while vaccinations are not mandatory in Canada, health crises like a Public Health Emergency of International Concern (PHEIC) [54], can warrant the legal suspension of civil liberties. Consequently, PHEIC level events require more careful treatment by public health authorities and government leaders.

Our results show consensus in Canadian opinion discourse regarding the health risk that those who do not comply with public health guidelines pose to the rest of society, given that pre-symptomatic and asymptomatic individuals could be spreading the virus. However, opinions were divided regarding potential solutions and responsibility. The message of prevention in Canada at the beginning of the pandemic was mostly premised on a system of trust, in which each Canadian was called to perform their moral duty and comply with public health guidelines. Despite a large segment of the population complying, some did not abide by the "new normal." Within our dataset, we found a variety of opinions regarding how to increase compliance with public health guidelines. These ranged from conservative voices arguing in favor of strong surveillance and the surrendering of privacy rights, to those who believe that blame should not be laid on individuals but on governments for not facilitating compliance, thus proposing a shared responsibility among government and citizens. This polarization is characteristic of opinion discourse [17], which is exacerbated by lack of previous experience and prevalent scientific uncertainty that come with the emergence of novel risks. Eventually, the federal government and some provincial governments resorted to fines as a way to enforce compliance with infection control guidelines. However, enforcement of these punitive measures was inconsistent.

Yet, while the fundamental public control measures were basically clear–self-isolate when sick or returning from travel, wear a facemask, and keep your distance–there was a myriad of confusing and conflicting guidance making it difficult for Canadians to stay on point. After the initial national lockdown in March/April, provinces with lower rates of COVID-19 began to relax restrictions in efforts to re-open the economy. Provincial leaders encouraged citizens to continue following public health guidance, while emphasizing individual and public health responsibilities for appropriate testing and contact tracing efforts to contain the spread of COVID-19 as bars, restaurants and the local economy re-opened. However, provincial and federal health authorities were slow to mandate the use of masks in public or indoor spaces. Notwithstanding, larger national chains (e.g. Wal-Mart) and smaller local businesses posted signs requiring that customers wear masks on entry [55, 56], and mask use became a requirement to access municipal offices and services such as public transit quite early in the pandemic, thereby contributing to its normalization. Despite this, private and public sector staff, often minimum wage employees, had little power to challenge a customer entering a business without a facemask or remind customers to distance while in the establishment. Compounding these issues were the challenges that provinces faced in ensuring public health and system capacity to follow through on its end of the responsibility scale. For example, many people, in their effort to 'do the right thing', faced long lines at testing sites and several days to know the result [57]. Likewise, due to rising case rates, public health became overwhelmed in its capacity to follow through on contact tracing efforts when a COVID positive result was known, such that in some jurisdictions, contract tracing was all but abandoned [58].

Despite the efforts to foster the desirable civic-minded moral behaviour of individuals through public health guidance, communication messaging created considerable 'gray zones'. Self-isolation messaging was rarely as clear as to go straight home and not make any stops along the way. Returning travellers were further confused by seemingly redundant and often unclear federal and provincial requirements for self-isolation. Further, encouraging

households to stick to their bubble, and minimize their number of contacts was regularly contradicted by provincial level messaging encouraging its residents to go out and support the local economy as part of re-start efforts. Eventually, many of the recommended public health guidance messaging which cajoled and appealed to Canadians to be their best-selves had to be abandoned in favour of stricter lock-down measures with heavy fines levied for individuals and businesses that broke the rules because case rates were rising exponentially and intensive-care treatment beds in hospitals were greatly exceeding capacity. Seasonal holiday celebrations of November and December 2020 (e.g. Diwali, Christmas, Chanukah, New Year's Eve) were dampened by messages to not celebrate with people outside their immediate household, leave their homes only for essentials, and to not engage in any unnecessary travel, in particular vacation travel. However, when it came to light that several provincial and federal government officials (Cabinet Ministers, backbenchers, and senior political staff) flouted stay-at-home recommendations by traveling to various national and international destinations over the holiday season [59], public outrage was high. While some Provincial Premiers reversed earlier decisions to not sanction these individuals [60, 61], others did not. It left some editorial writers wondering why citizens were being asked to view these transgressions with compassion and kindness, particularly given that non-governmental "idiots" who choose to defy guidance are blamed and shamed [62].

In this paper we have shown how Canadian news coverage of the COVID-19 pandemic contributed to the creation of a moral panic around use of facemasks, safe physical distancing, quarantine and isolation. Our results highlight the importance of clear public health communication to facilitate compliance with infection prevention norms and the need for governments to provide the material conditions to promote infection prevention behaviour and reduce the discrimination of certain groups and hostile reactions towards them. These are crucial aspects to consider as the COVID-19 vaccine program is starting in Canada, and those who cannot or choose not to be immunized could be the target of public condemnation and stigmatization.

We advance three recommendations for news coverage and risk communication of COVID-19. First, clear public health communication addressing issues causing confusion (i.e. who should isolate, when and how) would help increase compliance with infection prevention guidelines. These moments of confusion were expressed in both columns and letters to the editor, where contributors often also voiced frustration over lack of consistency in public health guidance. Communication efforts should highlight available community supports and resources to facilitate compliance with public health directives and lend support to initiatives such as paid sick leave that would allow people to comply with public health restrictions.

Second, the vilification in news coverage of those who violate public health guidelines can lead to the stigmatization of some groups, including those who cannot comply with certain guidelines, for example wearing a face mask or getting vaccinated, due to underlying medical conditions. Similarly, those who get infected with COVID-19 could also be vilified despite having complied with all the guidelines. By providing a more nuanced account of violations and compliance with preventive measures, journalists and editors can open up the discussion, addressing public concerns and social and structural barriers, instead of further polarizing superficial discussions over public compliance with guidelines.

Finally, opening up the discussion of compliance with public health directives to consider it not only a moral duty but also as a legal one would help normalize the fact that exceptional circumstances, such as a PHEIC level event, can trigger the legal suspension of some civil liberties. Journalists and editors could then frame legal restrictions as an acceptable course of action, stripping them of moral judgement, instead of proposing them as punishment for immoral behaviour. Nonetheless, COVID-19 has unearthed a series of issues related to

surveillance, civil liberties, community over individual level actions and responsibilities that warrant larger social discourse attention to better prepare us for the next pandemic. This would promote genuine engagement for what kinds of measures are appropriate relative to the threat being faced, so that we do not again find ourselves facing difficult decisions during the next 'unprecedented crisis'.

## Author Contributions

**Conceptualization:** Gabriela Capurro, Michelle Driedger.

**Data curation:** Gabriela Capurro.

**Formal analysis:** Gabriela Capurro.

**Funding acquisition:** Cynthia G. Jardine, Jordan Tustin, Michelle Driedger.

**Investigation:** Gabriela Capurro.

**Methodology:** Gabriela Capurro.

**Project administration:** Michelle Driedger.

**Software:** Gabriela Capurro.

**Supervision:** Michelle Driedger.

**Validation:** Michelle Driedger.

**Writing – original draft:** Gabriela Capurro.

**Writing – review & editing:** Cynthia G. Jardine, Jordan Tustin, Michelle Driedger.

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
