## [Decision Letter · Decision Letter 0]

2 Aug 2021

PONE-D-21-04611

“Good citizens” and “covidiots”: Moral panic in Canadian opinion discourse of COVID-19

PLOS ONE

Dear Dr. Capurro,

Thank you for submitting your manuscript to PLOS ONE. After careful consideration, we feel that it has merit but does not fully meet PLOS ONE’s publication criteria as it currently stands. Therefore, we invite you to submit a revised version of the manuscript that addresses the points raised during the review process.

The reviewer has recommended that you make minor revisions to the manuscript. Please attend to these suggestions and where you don't agree, provide reasons for that. Then return the revised manuscript to us as advised in this letter.

We look forward to receiving your revised manuscript.

Kind regards,

Martin Chtolongo Simuunza, PhD

Academic Editor

PLOS ONE

Journal Requirements:

Additional Editor Comments (if provided):

Reviewers' comments:

Reviewer's Responses to Questions

**Comments to the Author**

1. Is the manuscript technically sound, and do the data support the conclusions?

Reviewer #1: Yes

2. Has the statistical analysis been performed appropriately and rigorously? 

Reviewer #1: N/A

3. Have the authors made all data underlying the findings in their manuscript fully available?

Reviewer #1: Yes

4. Is the manuscript presented in an intelligible fashion and written in standard English?

Reviewer #1: Yes

5. Review Comments to the Author

Reviewer #1: Thank you for the opportunity to read this amazingly interesting paper. I am glad to see the authors take this topic on.

The paper is well written, easy to following and logically organized. My comments below are mere suggestions to make this already interesting and important paper even better.

The greatest (only) weakness of the paper is the treatment of moral panics. The authors claim that we are seeing a moral panic, or a sequence of them, about COVID noncompliance in opinion discourse. What the paper is really demonstrating is how opinion discourse presented noncompliance as a moral failure through shaming and the failure to ‘be in it together.’ The result was a sequence of calls for fines, quarantines, etc.

I think the three-fold focus on moral order/self-regulation, hostility, and punitive action not only makes sense but is demonstrated well in the findings section. But the link between the moral panic section and the findings one is not so strong.

I am aware that the authors have previously written on the relationship between moral regulation and moral panic. Yet none of this is discussed in the current paper. Too bad: it applies so well. The moral panic/regulation literature is full of discussion about personal and collective responsibility, blame, shaming, and so on. In that literature, moral panic is conceptualized as a key sociological process that contributes towards the maintenance of moral order. In plain terms, the literature conceptualizes panic as a kind of emergency response to perceived breakdowns in moral regulation (understood as responsible self-control). It seems like this literature was made for COVID claimsmaking!

I would encourage the authors to deal with the current state of moral panic studies more analytically. The framework as it stands is more rhetorical and descriptive than analytical.

More than that, I wonder if the authors considered the merits of an elite engineered panic model? Grounds for this come out in the discussion. My own take is that the public health establishment did not adequately address COVID, with their mixed messaging about masks, aerosol, early risk in Canada, schools, reopenings, etc. In fact, this is all coming out today in the so-called third wave.

I wonder (somewhat rhetorically) what role official claimsmakers play(ed) in creating the conditions for panic in the first place? That is, it’s their mixed messages, changing views, and sometimes bizarre recommendations that foster opportunities to offload blame the snowbirds, the beach-going kids, etc. It seems to me that bourgeois families pack the highways and fill the ferries and campgrounds every chance they get. It seems to me that those suffering ‘covid fatigue’ find many chances to ‘break the rules’ when the covidiot public health people tell them they can travel. The flip flopping is painful in BC. Bonnie Henry and her team have a habit of changing their minds at the last minute (New Years Eve, Spin classes, recently restaurants). The March 21 messaging around the light at the end of the tunnel also provided insufficient preparation for the growing infection numbers we are seeing in April.

My point is that blaming kids, the fatigued, travelers, etc. might draw attention away from the biggest covidiots of all. John Horgan actually blamed young people in BC recently and told them not to blow it for the rest of us. As far as I know, Tofino was packed with middle-aged people over Easter. So were the ferries.

I would like to see the authors intensify their own critique of the public health officers and politicians in the discussion.

In short, the panic-as regulation framework seems to fit here very well. Not sure why the authors don’t use it.

Otherwise, I really enjoyed the paper.

One last interesting thing. Some restaurants in Vancouver, folk devils of sorts, refused to close after Bonnie Henry randomly closed them to indoor dining in late March. Yet BC residents can lift weights at a gym.

6. PLOS authors have the option to publish the peer review history of their article (what does this mean?). If published, this will include your full peer review and any attached files.

Reviewer #1: No

---

## [Author Response · Author response to Decision Letter 0]

25 Aug 2021

Comment 1: The greatest (only) weakness of the paper is the treatment of moral panics. The authors claim that we are seeing a moral panic, or a sequence of them, about COVID noncompliance in opinion discourse. What the paper is really demonstrating is how opinion discourse presented noncompliance as a moral failure through shaming and the failure to ‘be in it together.’ The result was a sequence of calls for fines, quarantines, etc.

I think the three-fold focus on moral order/self-regulation, hostility, and punitive action not only makes sense but is demonstrated well in the findings section. But the link between the moral panic section and the findings one is not so strong.

I am aware that the authors have previously written on the relationship between moral regulation and moral panic. Yet none of this is discussed in the current paper. Too bad: it applies so well. The moral panic/regulation literature is full of discussion about personal and collective responsibility, blame, shaming, and so on. In that literature, moral panic is conceptualized as a key sociological process that contributes towards the maintenance of moral order. In plain terms, the literature conceptualizes panic as a kind of emergency response to perceived breakdowns in moral regulation (understood as responsible self-control). It seems like this literature was made for COVID claimsmaking!

I would encourage the authors to deal with the current state of moral panic studies more analytically. The framework as it stands is more rhetorical and descriptive than analytical.

Response: We appreciate the suggestion to include the concept of moral regulation in our analytical framework. We have introduced and explained the concept of moral regulation in the section about moral panics, and we have used it subsequently in our discussion to explain how official discourses of moral regulation prompted a moral panic to be expressed in opinion discourse. 

Comment #2: More than that, I wonder if the authors considered the merits of an elite engineered panic model? Grounds for this come out in the discussion. My own take is that the public health establishment did not adequately address COVID, with their mixed messaging about masks, aerosol, early risk in Canada, schools, reopenings, etc. In fact, this is all coming out today in the so-called third wave.

I wonder (somewhat rhetorically) what role official claimsmakers play(ed) in creating the conditions for panic in the first place? That is, it’s their mixed messages, changing views, and sometimes bizarre recommendations that foster opportunities to offload blame the snowbirds, the beach-going kids, etc. It seems to me that bourgeois families pack the highways and fill the ferries and campgrounds every chance they get. It seems to me that those suffering ‘covid fatigue’ find many chances to ‘break the rules’ when the covidiot public health people tell them they can travel. The flip flopping is painful in BC. Bonnie Henry and her team have a habit of changing their minds at the last minute (New Years Eve, Spin classes, recently restaurants). The March 21 messaging around the light at the end of the tunnel also provided insufficient preparation for the growing infection numbers we are seeing in April.

My point is that blaming kids, the fatigued, travelers, etc. might draw attention away from the biggest covidiots of all. John Horgan actually blamed young people in BC recently and told them not to blow it for the rest of us. As far as I know, Tofino was packed with middle-aged people over Easter. So were the ferries.

I would like to see the authors intensify their own critique of the public health officers and politicians in the discussion.

In short, the panic-as regulation framework seems to fit here very well. Not sure why the authors don’t use it.

Response: Using the concept of moral regulation we have included in the discussion some lines regarding how official discourses shifted blame to the individual, which was then reproduced in opinion discourse. We also incorporated this idea to our recommendations, by suggesting reporters and commentators be more careful to not engage in this blame-shifting but instead keep their focus on risk managers and how they are facilitating or hindering risk prevention measures.

---

## [Decision Letter · Decision Letter 1]

15 Dec 2021

Moral panic about “covidiots” in Canadian newspaper coverage of COVID-19

PONE-D-21-04611R1

Dear Dr. Capurro,

We’re pleased to inform you that your manuscript has been judged scientifically suitable for publication and will be formally accepted for publication once it meets all outstanding technical requirements.

Kind regards,

Martin Chtolongo Simuunza, PhD

Academic Editor

PLOS ONE

Additional Editor Comments (optional):

Reviewers' comments:

Reviewer's Responses to Questions

**Comments to the Author**

1. If the authors have adequately addressed your comments raised in a previous round of review and you feel that this manuscript is now acceptable for publication, you may indicate that here to bypass the “Comments to the Author” section, enter your conflict of interest statement in the “Confidential to Editor” section, and submit your "Accept" recommendation.

Reviewer #1: All comments have been addressed

2. Is the manuscript technically sound, and do the data support the conclusions?

Reviewer #1: Yes

3. Has the statistical analysis been performed appropriately and rigorously? 

Reviewer #1: N/A

4. Have the authors made all data underlying the findings in their manuscript fully available?

Reviewer #1: Yes

5. Is the manuscript presented in an intelligible fashion and written in standard English?

Reviewer #1: Yes

6. Review Comments to the Author

Reviewer #1: (No Response)

7. PLOS authors have the option to publish the peer review history of their article (what does this mean?). If published, this will include your full peer review and any attached files.

Reviewer #1: No

---

## [Editor Report · Acceptance letter]

7 Jan 2022

PONE-D-21-04611R1 

Moral panic about “covidiots” in Canadian newspaper coverage of COVID-19 

Dear Dr. Capurro:

I'm pleased to inform you that your manuscript has been deemed suitable for publication in PLOS ONE. Congratulations! Your manuscript is now with our production department. 

Kind regards, 

on behalf of

Dr. Martin Chtolongo Simuunza 

Academic Editor

PLOS ONE